# Coral Lipidome: Molecular Species of Phospholipids, Glycolipids, Betaine Lipids, and Sphingophosphonolipids

**DOI:** 10.3390/md21060335

**Published:** 2023-05-30

**Authors:** Tatyana V. Sikorskaya

**Affiliations:** A.V. Zhirmunsky National Scientific Center of Marine Biology, Far Eastern Branch, Russian Academy of Sciences, ul. Palchevskogo 17, 690041 Vladivostok, Russia; miss.tatyanna@yandex.ru; Tel.: +7-423-2310905

**Keywords:** lipidomics, Cnidaria, plasma membrane, ceramideaminoethylphosphonate, Octocorallia, Hexacorallia, *Millepora*, gorgonian corals

## Abstract

Coral reefs are the most biodiversity-rich ecosystems in the world’s oceans. Coral establishes complex interactions with various microorganisms that constitute an important part of the coral holobiont. The best-known coral endosymbionts are Symbiodiniaceae dinoflagellates. Each member of the coral microbiome contributes to its total lipidome, which integrates many molecular species. The present study summarizes available information on the molecular species of the plasma membrane lipids of the coral host and its dinoflagellates (phosphatidylcholine (PC), phosphatidylethanolamine (PE), phosphatidylserine (PS), phosphatidylinositol (PI), ceramideaminoethylphosphonate, and diacylglyceryl-3-*O*-carboxyhydroxymethylcholine), and the thylakoid membrane lipids of dinoflagellates (phosphatidylglycerol (PG) and glycolipids). Alkyl chains of PC and PE molecular species differ between tropical and cold-water coral species, and features of their acyl chains depend on the coral’s taxonomic position. PS and PI structural features are associated with the presence of an exoskeleton in the corals. The dinoflagellate thermosensitivity affects the profiles of PG and glycolipid molecular species, which can be modified by the coral host. Coral microbiome members, such as bacteria and fungi, can also be the source of the alkyl and acyl chains of coral membrane lipids. The lipidomics approach, providing broader and more detailed information about coral lipid composition, opens up new opportunities in the study of biochemistry and ecology of corals.

## 1. Introduction

The phylum Cnidaria comprises approximately 15,000 species (Figure 1). The best-studied cnidarians are corals, sea fans, and sea anemones [1]. Corals form coral reefs, which are the most biodiversity-rich ecosystems in the world’s oceans [1]. Lipidomic studies of corals include representatives of various taxonomic groups. Reef-building corals (Cnidaria: Anthozoa: Hexacorallia: Scleractinia), having a solid calcareous exoskeleton, constitute the structural basis of a reef [2,3], as well as symbiotic corals from the family Milleporidae (Cnidaria: Hydrozoa: Anthoathecata: Milleporidae) [4]. Soft corals (Cnidaria: Anthozoa: Octocorallia) with an internal skeleton composed of microscopic calcareous sclerites [5] have been considered the second most common group of macrobenthic animals after stony corals on many shallow-water reefs [6]. Alcyonacea is the most speciose order of Octocorallia [6], which includes gorgonian corals characterized by a keratin-like axial skeleton inside colonies [7]. 

Coral establishes complex interactions with a wide range of microorganisms that constitute an important part of the whole metaorganism, referred to as a holobiont [9]. The best-known coral symbionts are intracellular microalgae such as the dinoflagellates of the family Symbiodiniaceae. High densities of these obligate symbionts are found in the coral gastrodermis [9], where they produce nutrients for the host [10]. Under elevated sea water temperatures associated with the global warming effect, coral colonies lose Symbiodiniaceae [11]. This phenomenon, referred to as coral bleaching, results in the mortality of large coral reef ecosystems [12]. In addition to symbiotic dinoflagellates, the coral microbiome includes bacteria, Archaea, viruses [9], fungal communities [13,14], single-celled eukaryotic alveolates (ciliates, dinoflagellates, and chromerids), and apicomplexans (parasitic protists) [15].

Corals are rich in lipids, which constitute up to 23% of dry weight of coral tissues [16]. Lipids play an essential role in maintaining health and metabolism [17,18]; neutral lipids (triacylglycerols (TGs) and monoalkyldiacylglycerols) serve as a major reserve depot and energy source, while polar lipids, primarily phospholipids, perform a structural function and constitute the basis of cell membranes [19,20]. The lipids of tropical corals contain considerable amounts (13–50%) of several classes of polar lipids (PLs) including phosphatidylethanolamine, phosphatidylcholine, phosphatidylserine, phosphatidylinositol (PE, PC, PS, PI, respectively), and the sphingophosphonolipid ceramideaminoethylphosphonate (CAEP) [20]. Polar lipids, in addition to their main structural function, can be involved in various cellular processes. For example, PE is known to be involved in the regulation of inflammation, thrombosis, angiogenesis, and other important processes in higher animals [21]. Translocation of PS within the cellular membrane is reported as an indicator of apoptosis [22]. Some oxidized phospholipids are markers of pathological processes [21,23,24]. Rosic et al. (2015) suggested the involvement of PI in symbiotic interactions [25].

The lipids of the coral holobiont are a mixture of lipids of Symbiodiniaceae, the coral host, and other members of the coral microbiome. The typical cnidarian lipid classes, alkylacyl forms of PC, PE, PS, PI, and CAEP, are absent from Symbiodiniaceae lipids and can be considered markers of coral host tissues [26,27]. Several classes of glycolipids such as sulfoquinovosyldiacylglycerol (SQDG), mono- and digalactosyldiacylglycerol (MGDG and DGDG), and the phosphorus-containing class phosphatidylglycerol (PG) are essential components of membranes of the photosynthetic apparatus in plants and constitute the basis of *Symbiodiniaceae* lipids [28,29]. The plasma membrane of plant cells, in addition to diacyl forms of PC and PE, also contains betaine lipids (BLs). The latter lack phosphorus, and the ester bond present in PLs is replaced by a quaternary amine alcohol moiety with an ester bond at the *sn*-3 position [30]. The BL class 1,2-diacylglyceryl-3-*O*-carboxy-(hydroxymethyl)-choline (DGCC) has been identified in *Symbiodiniaceae* lipids [29,31,32].

Many studies of coral biochemistry were based on the integral lipid indices, e.g., the lipid content, composition of their fatty acids (FAs), or lipid class ratio. Total lipids or lipid classes are known to integrate several individual lipid molecules referred to as lipid molecular species, whereas total FAs are obtained by chemical degradation of the native lipid molecular species. The development of advanced physicochemical methods for lipid analysis in the past decade has allowed identification of structures of lipid molecules in their native form. The lipidome describes complex individual lipid species and varieties of their molecular forms in biological systems, from bacteria to mammals, while large-scale lipidomics studies consider the structure, function, interaction, and dynamics of lipid molecular species. All biological membranes contain a spectrum of lipid enantiomeric and diastereomeric species with diverse molecular shapes and differing in the length of the acyl chain, degree of unsaturation, head and backbone group composition, chirality, ionization, and chelating properties [33]. Lipids have no inherent catalytic activities and are not encoded by genes. However, complex lipidomes in multiple combinations can be remodeled to modulate collective membrane properties and determine the fluidity, supramolecular phase propensity, lateral pressure, and surface charge of the membrane bilayer. Lipidomics of marine organisms has rapidly developed since 2004 [1]. Compared to classical lipidology, the lipidomics approach provides more information about the lipid composition and allows more accurate quantitative analysis. 

The transition from classical integral lipid indicators to lipidome analysis can open up new opportunities in the study of corals’ biochemistry and ecology. Decoding the total lipidome of coral polyps will provide a basis for this kind of research. The present study aims to summarize the available information concerning the membrane lipid profiles of corals and the main members of their microbiome.

## 2. Coral Host Lipidome

Biological membranes (plasma membrane, thylakoid membrane, and membranes of intracellular organelles) are composed of a mixture of lipids, each with distinctive biophysical properties. Lateral and transversal sorting of lipids can promote creation of domains inside the membrane through local modulation of the lipid phase [34]. Large negative-curvature lipids such as MGDG, PE, and PS tend to form the HII phase or cubic phase (inverted tubules), large positive-curvature lipids such as lyso-lipids form the non-lamellar hexagonal HI phase (micellar tubules), whereas small-curvature lipids such as DGDG, SQDG, PC, PG, and PI form the lamellar phase (bilayer) [34]. The plasma membrane is particularly important for determining cell shape and confers cells with mechanical robustness against extrinsic mechanical stresses [35]. The structure and mechanical properties of the plasma membrane are mainly determined by the glycerolipid bilayer [36].

The total lipidome of only a few corals has been described [37,38,39,40,41]. Profiles of membrane lipids were characterized from the reef-building corals *Seriatopora caliendrum* [42,43] and *Acropora cerealis* [37], the asymbiotic cold-water soft coral *Gersemia rubiformis* [44], several species of symbiotic tropical soft corals such as *Capnella* sp. [45], *Xenia* sp. [46], *Sinularia macropodia* [45], *Sinularia* sp. [40], *S. heterospiculata* [39], and *S. siaesensis* [41], and several species of tropical gorgonian corals: *Junceella fragilis*, *Dichotella* sp., *Menella* sp., and *Astrogorgia rubra* [47]. Membrane lipid molecular species were analyzed in two symbiotic tropical hydrocorals, *Millepora dichotoma* and *M. platyphylla*, and the asymbiotic cold-water hydrocoral *Allopora stejnegeri* [48,49]. The sea anemone *Aiptasia pallida* [50,51] and the zoantharian *Palythoa tuberculosa* [38] were also subjected to analyses of glycerophospholipid molecular species (Table 1).

### 2.1. Phosphatidylcholine and Phosphatidylethanolamine

The profile of the PL molecular species of each coral order is characterized by certain distinctive features. The major PC and PE molecular species of soft, reef-building corals and *Millepora* hydrocorals are in alkylacyl form [37,38,40,41,42,45,46,47,48,49]. However, in the tropical zoantharian *P. tuberculosa*, in the cold-water hydrocoral *A. stejnegeri*, and in the reef-building coral *S. caliendrum*, the proportion of alkylacyl forms of PC accounts for 45%, 40%, and 30% of total PC, respectively [38,43,48]. Ether lipids (alkylacyl PL forms) that contain alkyl moieties at the *sn*-1 position of the glycerol backbone have been detected only in animal tissues and anaerobic bacteria [57,58]. The alkylacyl forms of PC and PE, with their source being, presumably, the coral host, are considered in this section. In the present study, the alkylacyl PL molecular species (PC, PE, PS, and PI) of different corals were grouped based on the acyl and alkyl chains of PL molecules. For visualization of lipidome differences, a cluster analysis and a principal component analysis (PCA) of PL compositions were carried out using the R statistical software (Figure 2). 

*Acropora cerealis* and the zoantharian *P. tuberculosa* contained a large amount of PC with both C_20_ and C_22_ PUFAs, e.g., PC 16:0alk/20:4, 16:0alk/22:5, and 18:0alk/20:4 [37,38] (Figure 2a). In *Sinularia*, *Capnella*, and *Xenia* species, almost all PC molecular species contained C_20_ PUFAs (16:0alk/20:4 and 18:0alk/20:4), which indicates a significant difference in the pathways of PC biosynthesis in octocorals and hexacorals [39,40,41,45] (Figure 2a). Due to a large number of PC molecules with C_20_ PUFAs, these corals formed a separate subcluster (Figure 2b). The highest content of C_16_ and C_18_ PUFAs (16:0alk/16:2, 16:0alk/18:2, and 18:0alk/18:3) was observed in *Sinularia* species, which led to their clustering (Figure 2a,b). In *Millepora* species (hydrocorals), the high level of PC molecular species with 22:5*n*-6 and 22:6*n*-3 (16:0alk/22:5, 16:0alk/22:6, and 18:0alk/22:5) against the background of an extremely low level of PC molecular species with C_20_ PUFAs may be explained by the use of mainly C_22_ PUFAs for the synthesis of PE and PC [48,49]. Therefore, the *Millepora* species occupied a position isolated from other cnidarians. (Figure 2b). The coral clustering on the basis of the composition of the PC alkyl chains was different (Figure 2c,d). The cold-water corals *A. stejnegeri* and *G. rubiformis* formed a separate cluster due to an increased content of PC molecular species with monoenoic alkyl chains (16:1alk/20:4, 18:1alk/20:4, and 18:1alk/20:5), as well as a higher content of PC molecular species with long-chain alkyl chains (20:1alk/20:5 and 22:1alk/20:4). It is known that the length and degree of unsaturation of glycerolipids’ acyl and alkyl chains can influence the membrane viscosity [60,61]. Thus, under conditions of low ambient temperatures, cold-water corals synthesize PC molecular species with monoenoic and long-chained alkyl chains to maintain the vital fluidity of cell membranes.

On the basis of the composition of the PE alkyl chains, the corals were divided into two clusters (Figure 2e,f). *Sinularia* species, *Capnella* sp., and cold-water coral species were characterized by a high content of PE molecular species with unsaturated alkyl chains. As in the case of PC, the cold-water corals *A. stejnegeri* and *G. rubiformis* had a high level of PE molecular species with more unsaturated alkyl chains: 16:2alk/20:4, 18:2alk/20:4, 20:3alk/20:5, and 20:2alk/20:4. It is worth mentioning that *S. heterospiculata* also contained a higher level of PE molecular species with polyenoic alkyl chains: 20:3alk/20:0OH, 20:3alk/22:0OH, and 20:3alk/21:0OH. The major PE molecular species of all corals contained arachidonic acid, e.g., PE 18:1alk/20:4, 16:1alk/20:4, 18:0alk/20:4, and 20:1alk/20:4 [37,38,40,41,45,46,47,48], except for the *Millepora* species, whose major PE molecular species contained 22:5 and 22:6 PUFAs (more than 90% of total PE: 18:1alk/22:5, 16:1alk/22:5, 19:1alk/22:5, and 18:1alk/22:6) [48,49] (Figure 2g,h). In contrast to the FAs of hexacorals, those of octocorals and zoantharians contained C_24_ PUFAs [26,62] and, therefore, PE molecules with 24:6*n*-3 and 24:5*n*-6 PUFAs (e.g. 18:1alk/24:5 and 18:0alk/24:5) are a distinctive feature of soft corals [41,47], while PE molecular species with non-methylene-interrupted C_24_ PUFAs (e.g. 18:1alk/24:3) are characteristic of the zoantharian *P. tuberculosa* [38].

### 2.2. Phosphatidylserine and Phosphatidylinositol

The alkylacyl forms of PS with saturated and monoenoic alkyl chains, as well as saturated and monoenoic acyl chains of diacyl forms of PS molecules, were summed and clustered using the R statistical software (Figure 3a). The distribution of alkyl and acyl chains in the profile of PS molecular species differs from that of the most abundant membrane lipids, PC and PE. This is probably related to the specific molecular features of PS. As the major lipid with a net-negative charge, PS is, therefore, responsible for providing much of the inner leaflet’s charge density. A significant role of PS, then, is interacting with proteins in a non-specific charge-based manner to permit their appropriate localization within the cell [63]. The gorgonians *J. fragilis* and *Dichotella* sp. and the zoantharian *P. tuberculosa*, as well as the cold-water gorgonian *G. rubiformis*, were characterized by a high level of PS molecular species with monoenoic alkyl or acyl chains (18:1alk/22:4, 18:1alk/24:2, 18:1alk/24:6, 20:1/24:5, and 20:1alk/24:5), in contrast to other cnidarians including the cold-water hydrocoral *A. stejnegeri*.

The alkylacyl forms of PS with different FAs, as well as different FAs of diacyl PS forms (paired monoenoic and saturated acyl chains), were summed and visualized by PCA using the R statistical software (Figure 3b). The major PS molecular species of the soft corals *Sinularia*, *Capnella,* and *Xenia* were in the alkylacyl form and amounted to >70% [39,40,41,45,46]. Most of them contained 24:5 and 24:6 PUFAs (18:0alk/24:5 and 18:0alk/24:6) which are known to be chemotaxonomic markers of soft corals [26,64]. In corals with a solid exoskeleton (*Acropora*, *Millepora*, and *Allopora*), only the diacyl forms of PS were identified (alkylacyl PS forms were in trace amounts) [37,48,50]. These PS molecular species contained 22:4, 22:5, and 22:6 PUFAs (18:0/22:4, 20:0/22:4, 18:0/22:5, 20:0/22:5, and 18:0/22:6). Tropical gorgonians, the cold-water gorgonian *G. rubiformis*, characterized by a keratin-like axial skeleton, and the zoantharian *P. tuberculosa,* with an internal skeleton composed of microscopic calcareous sclerites, occupied an “intermediate position” and contained 50–70% of alkylacyl PS forms [38,44,47]. The diacyl forms of PS molecular species of *P. tuberculosa* contained C_20_ and C_22_ PUFAs (18:0/22:4, 18:0/22:5, and 18:0/20:3), while alkylacyl PS forms additionally contained C_24_ PUFAs (18:1alk/24:2, 18:1alk/24:3, and 18:1alk/24:4). Ether-linked alkyl chains in phospholipids permit tighter packing of phospholipids in the membrane [57], whereas the length of the alkyl and acyl group of PL molecules strongly influences the thickness of lipid bilayer, which is critical for proper functioning of membranes [65]. Therefore, the matching of alkyl to longer acyl chains (24:5 and 24:6) in the PS molecule is likely to be a certain compensatory mechanism responsible for the physical properties of the membrane. 

The alkylacyl forms of PI with different FAs, as well as different FAs of diacyl forms of PI molecular species (paired monoenoic and saturated acyl chains), were summed and visualized by 3D Scatterplot (Figure 3c,d). All studied corals mainly contained more than 80% diacyl forms of PI molecular species. The profiles of the PI molecular species of the octocorals were very similar (18:0/24:5, 18:0/24:6, 18:0/22:4, 19:0/24:5, and 18:0alk/20:4), but the *Xenia* corals and *J. fragilis* showed the smallest proportion of PI molecular species with C_24_ PUFAs, which led to their separation into a cluster common with solid corals (Figure 3c). As in the case of PC, PE, and PS, the cold-water coral *G. rubiformis* was distinguished by an increased content of monoenoic alkyl or acyl chains (20:1/16:2, 20:1/20:4, and 20:1/20:5) (Figure 3d).

### 2.3. Ceramideaminoethylphosphonate

The sphingophosphonolipid CAEP is a typical structural lipid class of marine invertebrates [66]. Like glycerophospholipids, CAEP is one of the major membrane lipid classes of cnidarians. In some cases, it has a content comparable with the major PC and PE classes. The presence of a C-P bond in the aminoethylphosphonate of CAEP determines the resistance of CAEP to hydrolytic enzymes [67]. The function of phosphonolipids is poorly understood. It is likely that the CAEP resistance to hydrolysis maintains the vitality of coral endosymbiotic dinoflagellates when incorporated into gastrodermal coral cells [50]. In octocorals, as well as in hydrocorals *M. dichotoma* and *M. platyphylla*, CAEP molecular species with palmitic acid in the acyl moiety and 18 carbon atoms in the sphingosine base (18:3b/16:0, 18:2b/16:0, 18:1b/16:0, and 18:0b/16:0) were found [39,40,41,45,46,49]. CAEP molecular species with a unique triene type of sphingoid base (19:3) are an abundant sphingolipid in marine mollusks [68]. CAEP molecular species with this sphingoid base were identified in gorgonians and hydrocorals (19:3b/14:0, 19:3b/15:0, and 19:3b/16:0) [47,48,49]. The cold-water gorgonian *G. rubiformis* was characterized by a higher content of CAEP molecular species with 20 and 22 carbon atoms in the sphingoid base (22:3b/16:0, 22:2b/16:0, 20:3b/16:0, and 20:2b/16:0) [44]. The CAEP molecular species with a hydroxyl in the acyl chain (18:2b/16:0-OH) accounted for 6.2 and 17.4% of total CAEP molecular species in the soft corals *S. siaesensis* and *S. heterospiculata*, respectively. In the reef-building coral *A. cerealis*, higher contents of CAEP molecular species with hydroxyl both in the acyl chain and in the sphingoid base were found, e.g., 18:2b(OH)/16:0 and 18:3b(OH)/16:0(OH) were the major ones and amounted to more than 50% of total CAEP molecular species [37]. In the zoantharian *P. tuberculosa*, an N-methyl derivative of CAEP, ceramidemethylaminoethylphosphonate, was identified (18:2b/16:0, 18:2b/16:0-OH, and 19: 2b/16:0) that accounted for more than 60% of total phosphonolipids [38].

## 3. Lipidome of Symbiotic Dinoflagellates

The lipid compositions of the coral holobiont and the isolated symbiotic dinoflagellates differ sharply. The major lipid classes in algae are the thylakoid membrane lipids (MGDG, DGDG, SQDG, and PG), and plasma membrane lipids (PL and BL) [69]. Thylakoid membrane lipidomes of symbiotic dinoflagellates from cnidarians were studied [37,38,39,40,41,47,49,52,53] (Table 1). 

### 3.1. Plasma Membrane Lipids

The plasma membrane of algae contains PC, PE, PI, and BLs [69]. Ether (alkylacyl forms) lipids have not been detected in algae to date [57,58]. Therefore, diacyl forms of PC, PE, and PI molecular species of corals are considered in this section. In the asymbiotic coral species *G. rubiformis*, *Dichotella* sp., *Menella* sp., *A. rubra*, and *A. stejnegeri*, the coral host is the source of diacyl forms of PL molecular species. In the symbiotic coral species *S. heterospiculata*, *S. siaesensis*, *S. macropodia*, *Capnella* sp., *Xenia* sp., *J. fragilis*, *A. cerealis*, *S. caliendrum*, *M. platyphylla*, and *M. dichotoma*, the source of diacyl forms of PL molecular species can be both the coral host and symbiotic dinoflagellates. The marker PUFAs of Symbiodiniaceae are 16:4*n*-1, 18:3*n*-6, and 18:4*n*-3 [18,70]. Thus, PLs with these PUFAs can be synthesized by coral-symbiotic dinoflagellates. For example, PC 16:0/18:4 and PC 16:0/18:3 detected in the reef-building coral *S. caliendrum* can be synthesized by symbiotic dinoflagellates [42,43]. The high contents of 20:5*n*-3 and 22:6*n*-3 PUFAs were observed in lipids of Symbiodiniaceae cells isolated from corals [18,71]. However, PC 16:0/22:6, PC 18:0/22:6, PC 18:0/20:5, PC 16:0/20:5, PI 18:0/22:6, PI 18:0/20:5, and PI 16:0/22:6 were recorded from both symbiotic and asymbiotic corals. However, PC 22:6/22:6 was detected only in the symbiotic corals *S. caliendrum, M. platyphylla*, and *M. dichotoma*. Earlier, this PC was found in cultured symbiotic dinoflagellates [56]. Thus, the source of PC 22:6/22:6 is coral dinoflagellates.

The betaine lipid DGCC is one of the major structural lipids in the plasma membrane of coral-symbiotic dinoflagellates. Molecular species profiles of this lipid class were described from the symbiotic corals *Palythoa* sp., *A. cerealis*, *A. valida*, *S. heterospiculata*, *M. platyphylla*, and *M. dichotoma* [31,32,37,53]. The DGCC molecular species profile of *Symbiodinium microadriaticum* isolated from the jellyfish *Cassiopea xamachana* was identified [29]. *Acropora cerealis* from the coastal waters of Vietnam was characterized by DGCC 16:0/22:6 (38:6), 16:0/20:5 (36:5), and 18:0/28:7 (46:7) [36], whereas the DGCC profile of *A. valida* was different and characterized by a dominance of DGCC 38:6, 36:5, 44:12, and 42:11 for colonies that hosted *Cladocopium* C3 and for colonies that hosted *Durusdinium trenchii* [31]. The most abundant DGCC molecular species of *M. platyphylla* were 16:0/22:6 and 18:0/28:8, which differs from the results for *M. dichotoma* (DGCC 16:0/22:6, 18:0/28:7, and 16:0/18:4) [53]. The DGCC profile of the octocoral *S. heterospiculata* was characterized by a dominance of DGCC 16:0/22:6 (38:6), 40:9, and 16:0/20:5(36:5) [32]. The major DGCC species of the zoantharian *Palythoa* sp. were 16:0/18:4 and 16:0/22:6 [32].

The lyso-DGCC of coral-symbiotic dinoflagellates was also studied [31,55]. It was shown to be involved in coral bleaching. The increased accumulation of lyso-lipids may constitute a separate mechanism involved in the heat stress tolerance of the coral-symbiotic dinoflagellates *D. trenchii*.

### 3.2. Thylakoid Membrane Lipids

Dinoflagellates are a noteworthy example of algae with secondary plastids: in this case, with three envelope membranes and thylakoids inside [72]. The glycolipids SQDG, MGDG, and DGDG, and the phospholipid PG are essential components of membranes of the photosynthetic apparatus in plants. The anionic PG is considered a vital lipid, mainly for its role as a cofactor of photosystems. The anionic lipid SQDG with a sulfur-containing polar head interacts with photosynthetic proteins and some annexins [73]. Galactolipids, MGDG and DGDG, contain one and two galactose residues in their polar head, respectively [74]. In a thylakoid membrane, these lipids constitute a lipophilic matrix, which should allow the lateral diffusion of the photosystems [74,75]. In an organism under stable environmental conditions, the membrane lipidome in thylakoids is in a steady state (homeostasis), which is manifested as a constant ratio of thylakoid membrane lipids and their structure. Various lipid parameters such as degree of unsaturation and chain length determine the properties of the thylakoid membrane [60]. 

The Symbiodiniaceae exhibit a high genetic diversity with different physiological properties of the thylakoid membrane within and between species, resulting in the acclimation of their photosynthetic performance to various temperature and light conditions. For example, in contrast to the thermosensitive clades of coral-symbiotic dinoflagellates *Cladocopium* C3 and C1, the thermotolerant *D. trenchii* (clad D) is characterized by a higher degree of unsaturation of MGDG and DGDG and higher contents of saturated PG and SQDG [31,32,52]. The functional differences between symbiotic dinoflagellates are observed also at lower taxonomic levels, e.g., between species [76]. For example, the thermosensitive *Cladocopium* C3 is characterized by a very high SQDG/PG ratio, a DGDG/MGDG ratio <1, the lowest degree of galactolipid unsaturation, a higher content of SQDG with PUFAs, and a thinner thylakoid membrane, whereas other species of *Cladocopium* C3u and C71/C71a show thermotolerant lipidome features [52]. 

In the present study, the GL molecular species of different corals were grouped on the basis of acyl chain length and unsaturation. For visualization of lipidome differences, a heat map was composed, and a cluster analysis of GL compositions carried out (Figure 4). In addition to data on the GL molecular profile of whole coral colonies of *A. cerealis*, *Acropora* sp., *P. tuberculosa*, *J. fragilis*, *S. siaesensis*, *S. heterospiculata*, *S. flexibilis*, and *M. platyphylla* (two different data), data on the molecular profile of GL of the symbiotic dinoflagellates *Cladocopium* C1/C3 and *D. trenchii* isolated from *S. heterospiculata* (C(S)) and *P. tuberculosa* (D(P)), and also *Cladocopium* C1 (C(A)) and *D. trenchii* (D(A)) isolated from *A. valida* were taken into account (Table 1). It is worth noting C(A) were grouped with D(A), *A. cerealis*, and *Acropora* sp. and were not grouped with the same symbiont clade C(S) from another polyp host. This confirms that the host can influence the molecular species profile of the thylakoid membrane lipidome in a symbiotic dinoflagellate of coral.

The PL content in algae is much lower than the GL content. PG accounts for only 4% of membrane lipids of *Symbiodinium* [28]. Data on the PG molecular species of coral symbionts were provided in a few studies [31,52]. The most abundant PG molecular species in the *Cladocopium* C3 isolate from *A. valida* were 36:4 and 36:3, and those in the *D. trenchii* isolate from *A. valida* were 36:10 and 36:3. Dinoflagellates from *S. flexibilis* were characterized by a higher content of PG 16:2/20:2; dinoflagellates from *Acropora* sp., by higher contents of PG 16:2/20:2, 16:1/19:2, and 16:0/20:1; and dinoflagellates from *M. platyphylla*, by higher contents of PG 16:1/19:2, 16:1/18:2, 16:0/18:2, and 16:0/18:1. Earlier, the molecular species PG 16:1/19:2 and 16:2/20:2 were detected in the cultured symbiotic dinoflagellate species *Symbiodinium microadriaticum* (A1) and *C. goreaui* (C1) [56].

## 4. Lipidome of Other Members of Coral Holobiont

In shallow-water habitats with high levels of solar radiation, symbiosis with dinoflagellates promotes active coral growth and the formation of coral reefs. However, in low-light habitats, the role of dinoflagellates may become less important or altered, and other associations, e.g., with prokaryotes and endolithic eukaryotes, may play a more significant role than in shallow-water corals. FAs are successfully used in chemical systematics of various taxonomic groups including bacteria, fungi, macro- and microalgae, and corals [26,77]. Odd-numbered (straight- or normal-chain) FAs from 13:0 to 19:0 are found in an esterified form in the lipids of many bacterial species [77,78,79]. Additionally, branched FAs (i15:0, a15:0, i17:0, a17:0) are common constituents of bacterial lipids [80,81]. 

A coral bacterial community may include thousands of various bacterial species [82]. The concept of a core microbiome is based upon identifying the bacteria that are consistently present in the prokaryotic community, as opposed to those being only highly abundant [83,84]. A hypothesis has been advanced that associated microorganisms may be a partial “substitution” of photosynthetic symbiotic dinoflagellates as a source of organic carbon in asymbiotic coral species [85,86]. Previously, odd-numbered FAs as bacterial-specific markers were found in the lipids of different coral species [85,86,87]. The lipids of asymbiotic gorgonians contained a 4–8-fold higher level of PL molecular species with odd-numbered FAs than those of symbiotic coral [47]. In the corals considered in this report, some of the alkylacyl forms of PL molecular species were with odd-numbered alkyl or acyl chains such as PC 15:1alk/20:4, PC 17:0alk/20:4, PE 18:1alk/17:1, PE 19:1alk/20:4, PS 17:0alk/24:5, PS 19:0alk/22:4, PI 17:0/24:5, and PI 19:0/22:4. These PL molecular species can be of bacterial origin [88]. However, the high content of such molecular species is surprising. The total alkylacyl PC forms of *Dichotella* sp., *Menella* sp., and *A. rubra* constituted 16.06, 14.65, and 11.93% of the total PC, respectively [47]. PE 19:1alk/20:4 in *S. siaesensis*, *J*. *fragilis*, and *A. cerealis*, PE 18:1alk/17:1 and PE 16:1alk/17:1 in *P. tuberculosa*, and PE 19:1alk/22:5 in *Millepora* corals also had a higher content [37,38,41,47,49]. The bacterial origin of these molecular species or their odd-numbered FAs also calls into question the fact that the symbiotic dinoflagellates, both in hospite and cultured, contain odd-numbered molecular species of thylakoid membrane lipids: PG 16:1/19:2, DGDG 20:5/19:5, SQDG 16:0/17:0, and SQDG 16:0/17:2 [49,52,56].

Analyses of the microbiome of corals mainly focus on bacteria, although it may include several less-studied microorganisms including fungi [9,13,14,89], the most diverse and common of which are the *Aspergillus* and *Penicillium* genera of fungi [47,90,91,92]. Recently, hydroxylated C_18_, C_20_, C_22_, and C_24_ FAs were found in different species of fungi [93,94,95,96]. In the lipids of fungi, these FAs were detected only in TGs and ceramides and absent in PLs [96,97,98]. However, hydroxylated PL molecular species (PS 18:1alk/23:4(OH)_2_, PS 18:0alk/24:3(OH) and PS 19:1alk/24:3(OH), PS 18:0alk/18:1(OH)3 and PC 16:0alk/14:0(OH)) were detected in the lipids of gorgonian corals, which are associated with an advanced fungal community [47]. Thus, lipids are likely transported from the fungal community to the coral host for biosynthesizing hydroxylated PL molecular species. 

## 5. Conclusions

Thus, in this work, the accumulated information on the molecular species of membrane lipids of corals from various taxonomic groups was summarized. All corals are characterized by their specific profiles of lipid molecular species, which depend on various factors: coral taxonomic position, the environmental conditions of coral habitats, and Symbiodiniaceae species. In studied corals, the main lipids of the plasma membrane of the coral host are alkylacyl forms of PC and PE. They contain molecular species with odd-numbered alkyl chains and fatty acids, the source of which can be bacteria [77,79]. However, could bacteria be a source of these very abundant molecular species in corals? Symbiotic dinoflagellates also contribute to the entire pool of coral lipid molecular species. The structural lipids of the photosynthetic apparatus of coral-symbiotic dinoflagellates contain GL and PG. It has been shown that the ratio and molecular species profile of these lipids are associated with the thermosensitivity of symbiotic dinoflagellate species. Nevertheless, the influence of the coral host can be significant, as evidenced by the cluster analysis in the present study. In addition, the PG and SQDG of some coral species contain odd-numbered FAs [49,52,56]. PG is one of the major structural lipids of the bacterial plasma membrane, SQDG is also found in bacteria [99], and, therefore, these odd-numbered molecular species of PG and SQDG in the lipid extracts of corals can belong to bacteria. On the other hand, PG 16:1/19:2 is one of the major PG molecular species of the corals *Acropora* sp. and *M. platyphylla* [52]. In addition, this PG is detected in the lipids of various cultured symbiotic dinoflagellates [56]. This confirms that the source of the molecular species in the coral is definitely symbiotic dinoflagellates.

High-performance liquid chromatography and mass spectrometry have played a critical role in the development of lipidomics. The lipidomics approach provides more detailed information about the lipid composition of an organism and allows more accurate quantitative analysis. However, challenges remain at every level of the lipidomics experiment, rendering it difficult to compare data from different studies. To take lipidomics research to the next level, standardization of methods of lipidome analysis is needed.

## Figures and Tables

**Figure 1 marinedrugs-21-00335-f001:**
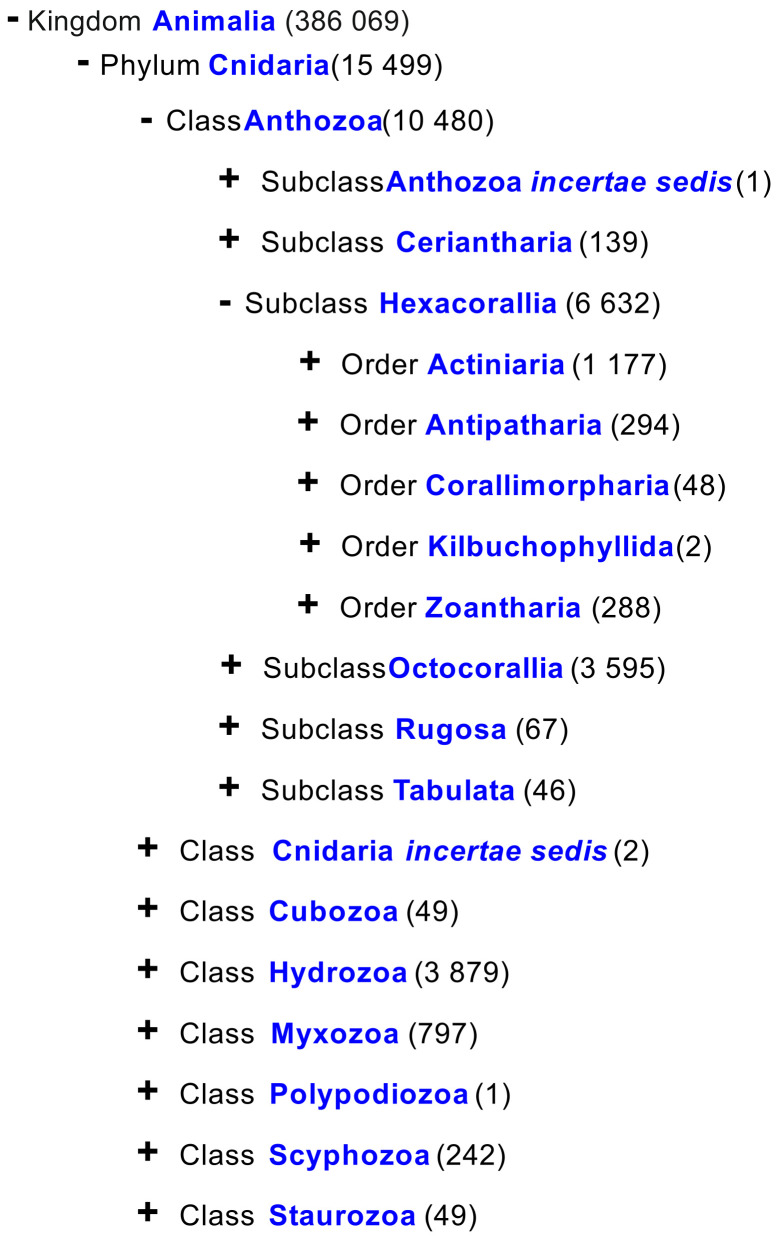
Taxon tree of the phylum Cnidaria [8].

**Figure 2 marinedrugs-21-00335-f002:**
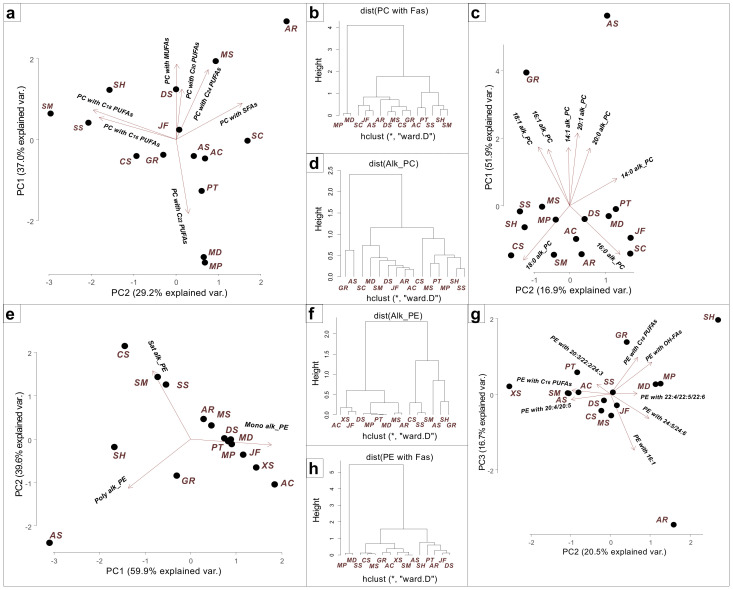
Lipidome features of corals. (**a**) A principal component analysis (PCA) of phosphatidylcholine (PC) molecular species composition with different fatty acids (FAs): saturated FAs (SFAs), monounsaturated FAs (MUFAs), C_16_ polyunsaturated FAs (PUFAs), C_18_ PUFAs, C_20_ PUFAs, C_22_ PUFAs, and C_24_ PUFAs; (**b**) clustering of these FAs. (**c**) PCA of PC molecular species with different alkyl chains: 14:0alk, 14:1alk, 16:0alk, 16:1alk, 18:0alk, 18:1alk, 20:0alk, and 20:1alk; (**d**) their clustering. (**e**) PCA of phosphatidylethanolamine (PE) molecular species with different saturation degree of alkyl chains: saturated (Sat_alk), monounsaturated (Mono_alk), and polyunsaturated; (**f**) their clustering. (**g**) PCA of PE molecular species with different FAs: C16 PUFAs, C18 PUFAs, 16:1 FA, hydroxylated FAs (OH-FAs), 20:4 and 20:5 FAs, 20:3, 22:2, and 24:3 FAs, 22:4, 22:5, and 22:6 FAs, 24:5 and 24:6 FAs; (**h**) their clustering. The preliminary data were arcsine-transformed prior to the PCA (eigenvalues of all components > 1) and cluster analysis (tree clustering, wards method, and Euclidean distances) [59]. The acronyms of coral species are as follows: SH—*Sinularia heterospiculata*; SS—*S. siaesensis*; SM—*S. macropodia*; CS—*Capnella* sp.; JF—*Junceella fragilis*; DS—*Dichotella* sp.; MS—*Menella* sp.; AR—*Astrogorgia rubra*; AC—*Acropora cerealis*; PT—*Palythoa tuberculosa*; SC—*Seriatopora caliendrum*; AS—*Allopora stejnegeri*; GR—*Gersemia rubiformis*; MP—*Millepora platyphylla*; MD—*M. dichotoma*.

**Figure 3 marinedrugs-21-00335-f003:**
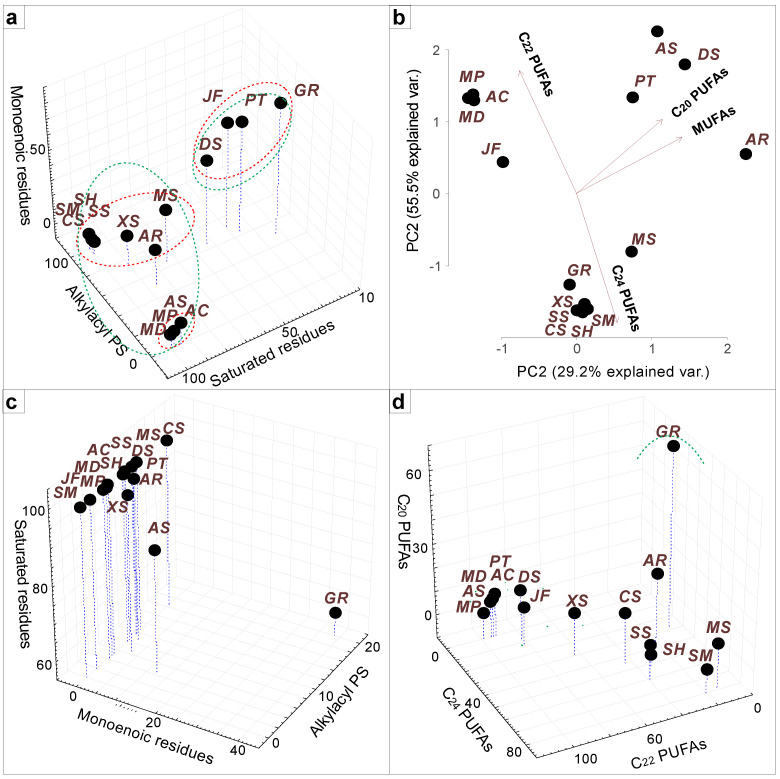
Lipidome features of corals. (**a**) Three-dimensional scatter plot of contents of total alkylacyl forms of phosphatidylserine (PS), all PS forms with saturated and monounsaturated alkyl and acyl chains. The dotted line outlines the following clusters: red, by content of total alkylacyl forms of PS; green, by saturation of PS. (**b**) A principal component analysis of PS molecular species composition with different fatty acids (FAs): monounsaturated FAs (MUFAs), C_20_ polyunsaturated FAs (PUFAs), C_22_ PUFAs, and C_24_ PUFAs. (**c**) Three-dimensional scatter plot of contents of total alkylacyl forms of phosphatidylinositol (PI), PI molecular species with saturated and monounsaturated alkyl and acyl chains. (**d**) Three-dimensional scatter plot of PI molecular species with different FAs: C_20_ PUFAs, C_22_ PUFAs, and C_24_ PUFAs, and their clustering. The dotted line outlines clusters. The preliminary data were arcsine-transformed prior to the PCA (eigenvalues of all components > 1) and cluster analysis (tree clustering, wards method, and Euclidean distances) [59]. The acronyms of coral species are as follows: SH—*Sinularia heterospiculata*; SS—*S. siaesensis*; SM—*S. macropodia*; CS—*Capnella* sp.; JF—*Junceella fragilis*; DS—*Dichotella* sp.; MS—*Menella* sp.; AR—*Astrogorgia rubra*; AC—*Acropora cerealis*; PT—*Palythoa tuberculosa*; XS—*Xenia* sp.; AS—*Allopora stejnegeri*; GR—*Gersemia rubiformis*; MP—*Millepora platyphylla*; MD—*M. dichotoma*.

**Figure 4 marinedrugs-21-00335-f004:**
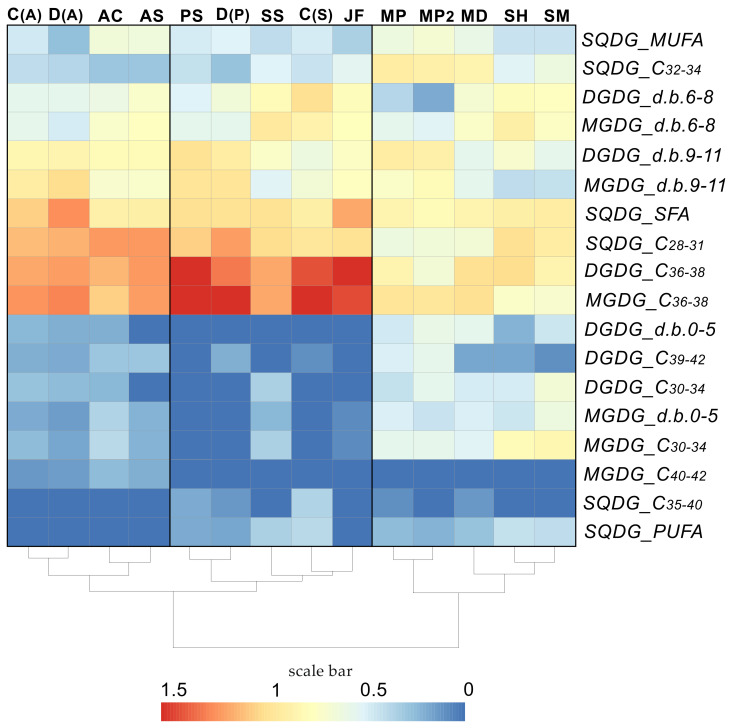
Lipidome features of corals. A heat map of glycolipid molecular species (sulfoquinovosyldiacylglycerol (SQDG), mono- and digalactosyldiacylglycerol (MGDG and DGDG)) grouped on the basis of acyl chain length (C_28–31_, C_30–34_, C_32–34_, C_35–40_, C_36–38_, C_39–42_, and C_40–42_) and fatty acid residues with different unsaturation degrees (saturated fatty acids (SFA), monounsaturated fatty acids (MUFA), polyunsaturated fatty acids (PUFA), FAs with 0–5 double bonds (d.b.), 6–8 d.b., and 9–11 d.b.); with clustering analyses (tree clustering, wards method, and Euclidean distances). The scale bar under the heat map(s) represents the arcsine-transformed relative abundance of lipid content in the samples [59]. The acronyms of coral or symbiont species are as follows: SH—*Sinularia heterospiculata*; SS—*S. siaesensis*; C(S)—*Cladocopium* C1/C3 from *S. heterospiculata*; D(S)—*Durusdinium trenchii* from *S. heterospiculata*; JF—*Junceella fragilis*; C(A)—*Cladocopium C1* from *Acropora valida*; D(A)—*D. trenchii* (D1) from *A. valida*; D(P)—*D. trenchii* (D1) and *Cladocopium* C1/C3 from *Palythoa tuberculosa*; AC—*A. cerealis*; PT—*P. tuberculosa*; MP—*Millepora platyphylla*; MD—*M. dichotoma*.

**Table 1 marinedrugs-21-00335-t001:** Corals studied by lipidomic approach. Plasma membrane lipids: phosphatidylethanolamine (PE); -choline (PC); -serine (PS); -inositol (PI); glycolipids (GL); ceramideaminoethylphosphonate (CAEP); betaine lipid (BL). Thylakoid membrane lipids: sulfoquinovosyldiacylglycerol (SQDG), mono- and digalactosyldiacylglycerol (MGDG and DGDG), and phosphatidylglycerol (PG).

**Class and Subclass of Cnidarians**	**Coral or Symbionts**	**Species Name**	**Membrane Lipid Class of Identified Molecular Species**	**Reference**
Anthozoa, Hexacorallia	Reef-building coral	*Acropora cerealis*	PC, PE, PS, PI, CAEP, GL, BL	[37]
Anthozoa, Hexacorallia	Reef-building coral	*Acropora* sp.	PG, GL, BL	[52]
Anthozoa, Hexacorallia	Reef-building coral	*Seriatopora caliendrum*	PC	[42,43]
Anthozoa, Hexacorallia	Zoantharian	*Palythoa tuberculosa*	PC, PE, PS, PI, CAEP, GL, BL	[38]
Anthozoa, Hexacorallia	Sea anemone	*Aiptasia pallida*	PC, PE, PS, PI, CAEP, SQDG	[50,51]
Anthozoa, Octocorallia	Soft coral	*Capnella* sp.	PC, PE, PS, PI	[45]
Anthozoa, Octocorallia	Soft coral	*Xenia* sp.	PC, PE, PS, PI, CAEP	[46]
Anthozoa, Octocorallia	Soft coral	*Sinularia heterospiculata*	PC, PE, PS, PI, CAEP, GL	[39,40]
Anthozoa, Octocorallia	Soft coral	*S. siaesensis*	PC, PE, PS, PI, CAEP, GL	[41]
Anthozoa, Octocorallia	Soft coral	*S. macropodia*	PC, PE, PS, PI	[45]
Anthozoa, Octocorallia	Gorgonian coral	*Junceella fragilis*	PC, PE, PS, PI, CAEP, GL	[47]
Anthozoa, Octocorallia	Gorgonian coral (asymbiotic)	*Dichotella* sp.	PC, PE, PS, PI, CAEP	[47]
Anthozoa, Octocorallia	Gorgonian coral (asymbiotic)	*Menella* sp.	PC, PE, PS, PI, CAEP	[47]
Anthozoa, Octocorallia	Gorgonian coral (asymbiotic)	*Astrogorgia rubra*	PC, PE, PS, PI, CAEP	[47]
Anthozoa, Octocorallia	Soft coral (asymbiotic, cold-water)	*Gersemia rubiformis*	PC, PE, PS, PI	[44]
Hydrozoa	Hydrocoral	*Millepora dichotoma*	PC, PE, PS, PI, CAEP, GL, BL	[48,49,53]
Hydrozoa	Hydrocoral	*M. platyphylla*	PC, PE, PS, PI, CAEP, GL, BL	[48,49,53]
Hydrozoa	Hydrocoral (asymbiotic, cold-water)	*Allopora stejnegeri*	PC, PE, PS, PI, CAEP	[48]
Dinoflagellates of the family Symbiodiniaceae	Symbionts from the jellyfish *Cassiopea xamachana*	*Symbiodinium microadriaticum*	GL, BL	[29]
Dinoflagellates of the family Symbiodiniaceae	Symbionts from the soft coral *S. heterospiculata*	*Cladocopium* C1/C3	GL, BL	[32]
Dinoflagellates of the family Symbiodiniaceae	Symbionts from the soft coral *S. heterospiculata*	*Durusdinium trenchii (D1)*	GL, BL	[32]
Dinoflagellates of the family Symbiodiniaceae	Symbionts from the reef-building coral *A. valida*	*Cladocopium* C1	GL, BL, PC	[31]
Dinoflagellates of the family Symbiodiniaceae	Symbionts from the reef-building coral *A. valida*	*D. trenchii (D1)*	GL, BL, PC	[31]
Dinoflagellates of the family Symbiodiniaceae	Symbionts from the soft coral *Capnella* sp.	–	MGDG, DGDG	[54]
Dinoflagellates of the family Symbiodiniaceae	Symbionts from the reef-building coral *Montipora capitata*	*Cladocopium* sp., *D. trenchii* (D1)	DGCC	[55]
Dinoflagellates of the family Symbiodiniaceae	Cultivated coral symbionts	*S. microadriaticum* (A1), *Cladocopium* C1, *Breviolum minutum* (B1)	GL, BL, PL	[56]

## Data Availability

Not applicable.

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
