# Peer review of "Coral Lipidome: Molecular Species of Phospholipids, Glycolipids, Betaine Lipids, and Sphingophosphonolipids"

_marinedrugs, 2023, doi:10.3390/md21060335_

Round 1
Reviewer 1 Report
This is an excellent and extensive review. It is very well written, and it was easy to follow, and a pleasure to read. This review will make an important contribution to the field of Cnidarian physiological ecology/comparative lipid biochemistry. I have only minor editorial suggestions that are as follows:
Line 54 - Please elaborate on the neutral lipid types which are used as a depot lipids. Are they triacylglycerols?
Line 68 - There is an incomplete sentence here which makes no sense.
Lines 108-110 - Please define what is meant by hexagonal I (HI) and hexagonal II (HII) lipid phases.
Table 1 - Please indicate that this is the lipodome of cellular membranes only. Please also clarify if these data are for plasma membranes or organelle membranes.
Author Response
Response to Reviewers
I would like to cordially thank the reviewers for their valuable comments. I have carefully read the comments and suggestions and made the relevant changes to the manuscript. Below are the point-by-point responses to the comments.
Reviewer: 1
- COMMENT: Line 54 - Please elaborate on the neutral lipid types which are used as a depot lipids. Are they triacylglycerols?
REPLY: I changed the sentence “Corals are rich in lipids, which make up to 23% of dry weight of coral tissues [16]. Lipids play an essential role in maintaining health and metabolism [17, 18]; neutral lipids (triacylglycerols and monoalkyldiacylglycerols) serve as a major reserve depot and energy source, while polar lipids, primarily phospholipids, perform a structural function and constitute the basis of cell membranes [19, 20].”
- COMMENT: Line 68 - There is an incomplete sentence here which makes no sense.
REPLY: I changed the sentence “The typical cnidarian lipid classes (alkylacyl forms of PC, PE, PS, PI and CAEP) are absent from Symbiodiniaceae lipids and can be considered markers of coral host tissues [26, 27].”
- COMMENT: Lines 108-110 - Please define what is meant by hexagonal I (HI) and hexagonal II (HII) lipid phases.
REPLY: I added this information in the manuscript “Large negative curvature lipids such as MGDG, PE, and PS tend to form the HII phase or cubic phase (inverted tubules,), large positive curvature lipids such as lyso-lipids form non-lamellar hexagonal HI phase (micellar tubules), whereas small curvature lipids such as DGDG, SQDG, PC, PG, and PI form lamellar phase (bilayer) [34].”
- COMMENT: Table 1 - Please indicate that this is the lipodome of cellular membranes only. Please also clarify if these data are for plasma membranes or organelle membranes.
REPLY: I added this information in the title of Table1.

Reviewer 2 Report
Comments
The review entitled “Сoral Lipidome: Molecular Species of Phospholipids, Glycolipids, Betaine lipid, and Sphingophosphonolipid” written by Tatyana V. Sikorskaya summarizes available information on molecular species of coral membrane lipids (phospholipids, glycolipids, betaine lipid, and sphingophosphonolipid). In general, the lipidomics approach provides more detailed information about the lipid composition of an organism and allows more accurate quantitative analysis. This paper is generally well-written. Thus, I suggest it can be accepted after revision.
Some points:
1. Key data are suggested presented in the abstract.
2. Please expand the discussion before conclusion.
3. Line 132, The major PC, PE molecular---The major PC and PE molecular
4. The styles of these references should be checked and revised according to the requirements of the journal, such as references 1, 6, 7, 9 etc.
Author Response
Response to Reviewers
I would like to cordially thank the reviewers for their valuable comments. I have carefully read the comments and suggestions and made the relevant changes to the manuscript. Below are the point-by-point responses to the comments.
Reviewer: 2
- COMMENT: Key data are suggested presented in the abstract.
REPLY: The abstract was changed.
- COMMENT: Please expand the discussion before conclusion.
REPLY: The discussion before conclusion was expand: “Analyses of the microbiome of corals mainly focus on bacteria, although it may include several less studied microorganisms including fungi [9, 13, 14, 89], the most diverse and common of which are Aspergillus and Penicillium genera of fungi [47, 90-92]. Recently, hydroxylated C18, C20,C22 and C24 FAs were found in different species of fungi [93-96]. These FAs detected in TG and ceramides and were absent in PL [96-98]. However, hydroxylated PL molecular species (PS 18:1alk/23:4(OH)2, PS 18:0alk/24:3(OH) and PS 19:1alk/24:3(OH), PS 18:0alk/18:1(OH)3 and PC 16:0alk/14:0(OH)) were presented in gorgonian corals which associated with an advanced fungal community [47]. Thus, lipids are likely transported from fungal community to the coral host for biosynthesizing hydroxylated PL molecular species”
And additional references (90-98) were added.
- COMMENT: Line 132, The major PC, PE molecular---The major PC and PE molecular
REPLY: I changed the sentence to:“The major PC and PE molecular species of soft, reef-building corals, and Millepora hydrocorals are in alkylacyl form [37, 38, 40-42, 45-49].”
- COMMENT: The styles of these references should be checked and revised according to the requirements of the journal, such as references 1, 6, 7, 9 etc.
REPLY: I checked and revised the references.
